# Associations between Respiratory Health Outcomes and Coal Mine Fire PM_2.5_ Smoke Exposure: A Cross-Sectional Study

**DOI:** 10.3390/ijerph16214262

**Published:** 2019-11-02

**Authors:** Amanda L. Johnson, Caroline X. Gao, Martine Dennekamp, Grant J. Williamson, David Brown, Matthew T. C. Carroll, Jillian F. Ikin, Anthony Del Monaco, Michael J. Abramson, Yuming Guo

**Affiliations:** 1Department of Epidemiology and Preventive Medicine, School of Public Health and Preventive Medicine, Monash University, Level 2, 553 St Kilda Road, Melbourne, VIC 3004, Australia; amanda.johnson@monash.edu (A.L.J.); caroline.gao@monash.edu (C.X.G.); Martine.Dennekamp@monash.edu (M.D.); David.Brown@monash.edu (D.B.); jill.blackman@monash.edu (J.F.I.); anthony.delmonaco@monash.edu (A.D.M.); Michael.Abramson@monash.edu (M.J.A.); 2School of Natural Sciences, University of Tasmania, Hobart, Tasmania 7001, Australia; Grant.Williamson@utas.edu.au; 3Monash Rural Health – Churchill, Monash University, Northways Rd, Churchill, VIC 3842, Australia; Matthew.Carroll@monash.edu

**Keywords:** fine particulates (PM_2.5_), surveys, cough, wheeze, sputum

## Abstract

In 2014, wildfires ignited a fire in the Morwell open cut coal mine, Australia, which burned for six weeks. This study examined associations between self-reported respiratory outcomes in adults and mine fire-related PM_2.5_ smoke exposure. Self-reported data were collected as part of the Hazelwood Health Study Adult Survey. Eligible participants were adult residents of Morwell. Mine fire-related PM_2.5_ concentrations were provided by the Commonwealth Scientific and Industrial Research Organisation Oceans & Atmosphere Flagship. Personalised mean 24-h and peak 12-h mine fire-related PM_2.5_ exposures were estimated for each participant. Data were analysed by multivariate logistic regression. There was some evidence of an association between respiratory outcomes and mine fire PM_2.5_ exposure. Chronic cough was associated with an odds ratio (OR) of 1.13 (95% confidence interval 1.03 to 1.23) per 10 μg/m^3^ increment in mean PM_2.5_ and 1.07 (1.02 to 1.12) per 100 μg/m^3^ increment in peak PM_2.5_. Current wheeze was associated with peak PM_2.5_, OR = 1.06 (1.02 to 1.11) and chronic phlegm with mean PM_2.5_ OR = 1.10 (1.00 to 1.20). Coal mine PM_2.5_ smoke exposure was associated with increased odds of experiencing cough, phlegm and wheeze. Males, participants 18–64 years, and those residing in homes constructed from non-brick/concrete materials or homes with tin/metal roofs had higher estimated ORs. These findings contribute to the formation of public health policy responses.

## 1. Introduction

Coal mine fire smoke contains multiple pollutants known to be harmful to human health, including particulate matter (PM), carbon monoxide, polycyclic aromatic hydrocarbons and benzene [1,2]. The chemical profile of pollutants released varies with geographical location, coal composition, meteorology and combustion conditions [3]. Of these pollutants, fine PM_2.5_, being particulate matter with an aerodynamic diameter <2.5 µm, has been identified as the most harmful to human health [4]. PM_2.5_ are a mixture of solid and liquid particles created either as primary particles emitted directly into the atmosphere or secondary particles formed in the atmosphere from pollutant gases.

Coal mine fires are often ignited by surface fires (e.g., wildfire), spontaneous combustion or mine-related activities. These fires can be difficult to extinguish and multiple coal mine fires are currently active across the world’s coal-bearing nations [1,2]. For example, in Australia, the “Burning Mountain” coal fire has been active for over 6000 years [5] and in India, the Iharia Coal fields fire has been active for over a century [6].

Limited research is available regarding the detailed emission characteristics of open cut coal mine fires and their possible effects on human health [3,7]. To date, research regarding coal mine fire emission characteristics has primarily focussed on underground mine fires in South Africa and the US, using measurements of exhaust fumes taken at vent surfaces [1,7,8]. Furthermore, research specific to the brown lignite coal involved in this study has focused on combustion in industrial furnaces, which takes place at higher temperatures relative to those in open cut mines and therefore, is more complete. As a result, the emission characteristics and potential health effects of coal burnt in underground mines or industrial furnaces would be expected to differ from those of coal burnt in an open cut mine fire [1,8].

Wildfires, like coal mine fires, involve the incomplete combustion of solid hydrocarbon-based fuel sources and are thought to have a broadly similar spectrum of chemical emissions [7]. Therefore, it is plausible that the organ systems affected by both sources of fire would be similar, although wildfires are often of a shorter duration than mine fires. Multiple studies have found associations between exposure to wildfire PM and adverse respiratory outcomes, including exacerbations of asthma and COPD, bronchitis and pneumonia [9,10,11,12,13,14,15]. Mine fire PM_2.5_ has been associated with increased medication dispensing for respiratory health conditions [16] and studies have found that populations living in coal mining communities are at increased risk of respiratory health conditions, even in the absence of fire [17]. 

In February 2014, local wildfires ignited a fire in the Morwell open-cut coal mine adjacent to the Hazelwood power station and the town of Morwell, in south-eastern Victoria, Australia (Figure 1). The mine fire burned for approximately six weeks. Hourly mine fire-related PM_2.5_ concentrations were estimated to have reached 3700 µg/m^3^ [8] during the initial phase of the fire. The daily average National Environment Protection Measure (NEPM) standard of 25 µg/m^3^ was breached on 27 days during February and March 2014 in the Morwell township [8,18]. In response to community concerns following the fire, the Hazelwood Health Study (HHS) was established to investigate the potential long-term health effects of the mine fire on the local population. The current analysis draws upon the Adult Survey component of the HHS to examine whether there was an association between mine fire-related PM_2.5_ smoke exposure and self-reported respiratory health outcomes for adult residents of Morwell, approximately 2.5 years after the mine fire.

## 2. Materials and Methods 

### 2.1. Study Design and Setting

We conducted a cross-sectional study of self-reported respiratory outcomes following exposure to coal mine fire-related PM_2.5_. The methods have been published elsewhere [19,20]. In brief, the study was conducted in the Latrobe Valley, south-eastern Victoria, Australia. The exposure zone comprised five Statistical Area 2 (SA2) districts: Morwell, Yallourn North, Moe, Churchill and Traralgon [21] (Figure 1). The region was semi-rural and the mine was located on the south-western boundary of the Morwell township. The exposure period was defined as 9 February–31 March 2014 and the recruitment period was May 2016–February 2017. 

### 2.2. Participants

Participants were residents of Morwell aged 18 years or older at the time of the mine fire. Eligible participants were identified using the Victorian Electoral Commission (VEC) roll and were invited to take part in the Adult Survey by mail to their last known address. Non-responders were followed up via telephone, mail and public advertisements. 

### 2.3. Variables

Data was collected via a self-report survey, over the phone, online, or on paper. Variables included participants’ demographic and socioeconomic indicators (age, gender, marital status, education level and employment status). A modified version of the European Community Respiratory Health Survey [22] was used to identify respiratory symptoms in the previous 12 months and respiratory conditions since 2014. 

Participants reported any paid jobs held for at least six months, which may have involved exposure to dust, fumes, smoke, gas vapour or mist. Participants also reported employment in the Latrobe Valley coal mines or power stations, any paid or volunteer positions with the emergency services and, specifically, fire-fighting in the Hazelwood mine fire Controlled Area. Based upon responses, participants were divided into three occupational exposure categories: not exposed; coal mine or power station; exposed but not coal mine power station. Respondents who had ever smoked at least 100 cigarettes or a similar amount of tobacco in their lifetime were defined as *smokers* as per the World Health Organization (WHO) definition [23] and categorised as *current* or *former*. 

Participants also identified the year of construction of their residence, the main building material, type of roofing and any use of air conditioning during the exposure period. A time-location diary was completed detailing participants’ day and night residential, work and any relocation addresses during the 51 day/night exposure period.

Mine fire-related PM_2.5_ concentrations were retrospectively modelled by the Australian Commonwealth Scientific and Industrial Research Organisation (CSIRO) Oceans & Atmosphere Flagship, using The Air Pollution Model (TAPM v4.0.5) [24,25], combined with a chemical transport model (CTM). TAPM is a fine-scale air pollution model without chemistry and CTM is a broader scale model that includes atmospheric chemistry. Both are driven by a separate downscaled weather model, the Conformal Cubic Atmospheric Model (CCAM) [26]. Modelled data were used due to a paucity of air quality monitoring at the time of the fire [1,3,8]. In particular, no data were available for the first 10 days of the fire in the residential areas of Morwell closest to the mine. 

Modelling was conducted at two spatial scales. High-resolution near-field modelling with a spatial resolution of 100 m was undertaken for the 10 × 10 km^2^ area around the coal mine and the town of Morwell. Maps of area burned and estimated emission factors were used to model hourly PM_2.5_ emission rates. Emissions were dispersed using TAPM driven by CCAM-downscaled meteorology and were treated as tracer species, i.e., no secondary chemical reactions. Verification of the hourly modelled PM_2.5_ concentrations, compared to available measured observations (for the period 20 February–28 March 2014) found correlation coefficients of *r* = 0.57 close to the mine and *r* = 0.37, 3 km downwind of the mine [8]. Modelled estimates were found to predict the correct magnitude, but the timing of the maxima was not always synchronised with measured observations [8]. However, for this analysis, hourly model output was aggregated to daily mean concentrations, which would have improved the temporal reliability of the model.

Regional modelling was conducted using the CTM driven by the CCAM-downscaled meteorology and incorporated full chemistry simulations. The model incorporated a set of nested grids with resolutions ranging from 1 km close to Morwell, to 80 km at the national scale. Two model runs were performed. The first included only background sources of PM_2.5_ which incorporated all anthropogenic sources and active wildfires. The second also included PM_2.5_ emissions from the coal mine fire. Subtracting one run from the other allowed mine fire-related concentrations to be calculated. A more detailed description of the modelling approach can be found in Emmerson [8]. 

As the coal mine fire burned continually over a number of weeks, the model not only simulated diffusion after the fire but innovative efforts were also made to account for the continuity of the fire. These included the use of aerial linescan imagery to map the area of the mine burning each hour, the McArthur Forest Fire Danger Index [27] to modify the emission rates over time in response to changing weather conditions, and a dynamic plume rise model, described in Emmerson [8] to vary the height of the emissions column in response to wind and area burning.

Personalised mean 24-h and peak 12-h mine fire-related PM_2.5_ exposure metrics were calculated for each participant based on the addresses listed in their time-location diaries. The Morwell SA2 consisted of 36 SA1s (Figure 2) and exposure was assigned to Morwell addresses at that scale. Exposure was assigned to Churchill, Moe, Yallourn North or Traralgon addresses at the SA2 level (Figure 1). The temporal scale was 12 hours: 6:00–18:00 defined as day time exposure and 18:00–6:00 defined as night exposure. For each participant, their 51-day and 51-night PM_2.5_ concentrations were averaged to obtain their cumulative mean 24-h exposure metric. Their peak 12-h PM_2.5_ concentration metric was the maximum SA1 or SA2 area level concentration assigned to the addresses recorded in their diary.

### 2.4. Statistical Methods

Associations between mine fire smoke exposure and respiratory outcomes were investigated for mean and peak PM_2.5_ concentrations using multivariate logistic regression. Analyses were conducted using weighted methods of estimation to reduce the possibility of participation bias. The final multivariate models included exposure variables (mean or peak PM_2.5_ exposure) and common confounders of the respiratory outcomes, namely, age, gender, employment, education, marriage status, smoking, employment exposure, roof materials, building materials, pre-existing asthma and pre-existing chronic obstructive pulmonary disease (COPD). Interactions between PM_2.5_ exposure and both housing and roofing type were tested to rule out the impact of smoke ingress on the true exposure of participants. 

Missing data for roofing type and building materials were completed using publicly available real-estate websites and satellite images. Missing observations for remaining variables ranged between 0.06% and 1.3% (Appendix A) and were accounted for using multiple imputation by chained equations (MICE), with 20 iterations [28,29]. 

Overall effects were measured using the odds ratio (OR) for self-reported respiratory outcomes and 95% confidence intervals (CI), associated with a 10 μg/m^3^ increase in mean 24-h and a 100 μg/m^3^ increase in peak 12-h mine fire-related PM_2.5_ smoke concentration. Stratified analyses were also conducted to evaluate possible age (18–64 vs. 65+ years) and sex (males vs. females) differences. Statistical analysis was conducted using Stata 15 [30].

### 2.5. Ethics Approval

The protocol for the Adult Survey was approved by the Monash University Human Research Ethics Committee (Project number 6066). Participants provided informed consent. 

## 3. Results

Recruitment results and bias assessment are detailed elsewhere [19,20]. In brief, 9013 Morwell residents were identified by the VEC roll as eligible. Of those, 3037 (34%) participated, as well as 59 Morwell residents who had not been listed on the roll (*n* = 3096). An assessment of sampling bias found an over-representation of women and older people and differences in smoking patterns. To account for potential bias, the analysis was conducted using weighted methods of estimation (Table 1). Recruitment rates were marginally higher for SA1s closest to the mine (Appendix A). 

Mean 24-h mine fire-related PM_2.5_ concentrations experienced by participants ranged from 0 to 56 µg/m^3^ with a median of 11 µg/m^3^ and peak 12-h concentrations ranged from 0 to 879 µg/m^3^ with a median of 132 µg/m^3^ (Figure 3). Peak and mean exposure concentrations had similar distributions whilst peak concentrations had more high-value outliers. The Morwell SA1s closest to the mine experienced the highest concentrations and were also the most highly populated residential areas (Figure 1 and Figure 2). 

Potential confounding and outcome variables across tertile categories of mean 24-h PM_2.5_ exposure are shown in Table 1 and Table 2. Exposure for the period 9 February–31 March 2014 was categorised as: Low: 0-8.6 µg/m³, Medium: >8.6–14.1 µg/m³ and High: >14.1–56.0 µg/m³. Participant age, marital status, smoking status, housing materials and roofing materials varied across exposure categories, which highlighted the need for controlling these variables in the multivariate analysis (Table 1). 

The multivariate analysis showed a pattern of increasing respiratory symptoms with increasing PM_2.5_, however, CIs were often wide and statistical significance was not achieved (Figure 4, Appendix A). For each outcome, the estimated OR for a 10 μg/m^3^ increment in mean PM_2.5_ was similar to the OR for a 100 μg/m^3^ increment in peak PM_2.5_, and CIs were wider for mean concentrations. The strongest associations were for chronic cough, with ORs of 1.13 (95%CI 1.03 to 1.23; *p*-value 0.007) per 10 μg/m^3^ of mean PM_2.5_ and 1.07 (1.02 to 1.12; 0.004) per 100 μg/m^3^ of peak PM_2.5_. Current wheeze was associated with peak PM_2.5_, OR = 1.06 (1.02 to 1.11; 0.004) and chronic phlegm was associated with mean PM_2.5_, OR = 1.10 (1.00 to 1.20; 0.052). Those with pre-existing asthma or COPD reported worse respiratory symptoms than those without (Table 1), but no interaction with exposure was identified. A comparison using MICE and a complete case analysis yielded essentially the same results. 

The sex stratified analyses suggest that estimated ORs were generally higher for males compared with females (Figure 5, Appendix A). The highest ORs were observed in men with asthma since 2014, OR 1.58 (1.10 to 2.29; 0.014) for mean PM_2.5_ and 1.43 (1.14 to 1.78; 0.002) for peak PM_2.5_, with little evidence in women. Among men, the ORs for chronic cough were estimated as 1.20 (1.05 to 1.37; 0.007) for mean PM_2.5_ and 1.07 (1.00–1.14; 0.050) for peak PM_2.5_ and current wheeze 1.10 (1.03 to 1.17; 0.004) for peak PM_2.5_. Amongst females, the highest ORs were for chronic cough, OR 1.06 (1.00 to 1.12; 0.051) for peak PM_2.5_. Age stratification showed higher estimated ORs for participants aged 18–64 years, compared with those 65 and over (Figure 6, Appendix A). For those 18–64, the highest ORs were observed for chronic cough, OR 1.17 (1.04 to 1.30; 0.008), and chronic phlegm, OR 1.14 (1.02 to 1.28; 0.023), both for mean PM_2.5_. For those 65 and over, all CIs incorporated the value one.

A protective effect of brick/concrete housing materials was identified in the multivariate regression models for both mean and peak PM_2.5_, for wheeze, cough and nocturnal and resting shortness of breath symptoms (Appendix A). Two-way interaction analyses of home building materials and PM_2.5_ concentrations, found associations between nocturnal and resting shortness of breath outcomes and the interaction variable for mean PM_2.5_ (Appendix A). Estimated ORs between PM_2.5_ and respiratory symptoms were higher for participants with non-brick/concrete housing compared with those with brick/concrete housing (Appendix A). For participants with metal/tin roofing materials, the multivariate regression models estimated increased odds for both mean and peak PM_2.5_, for nasal symptoms and phlegm (Appendix A). No interaction was identified between roofing materials and PM_2.5_ concentrations. 

## 4. Discussion

To the best of our knowledge, this is the first study to examine self-reported respiratory symptoms associated with smoke exposure from a coal mine fire. Our findings showed some evidence of an association between mine fire-related PM_2.5_ smoke exposure and self-reported respiratory outcomes collected about 2.5 years after the fire. The strongest relationship observed was between chronic cough and mean PM_2.5_ exposure. Chronic cough and current wheeze were associated with peak exposure, and chronic phlegm with mean exposure. Males, participants aged 18–64 years, and those residing in homes constructed from non-brick/concrete materials or homes with tin/metal roofs had higher estimated ORs.

Our analysis builds on the HHS Adult Survey volume 2 report [19], which found an association between chronic cough and tertile categories of participants’ mean 24-h mine fire-related PM_2.5_. In our analysis, building and roof material types were included as additional predictor variables, age and gender stratification were undertaken, peak 12-h exposure was examined, and exposure was analysed as a continuous variable. It is possible that these factors improved the statistical power of the model, facilitating the identification of additional associations between respiratory outcomes and mine fire-related PM_2.5_ smoke exposure.

Direct comparisons between our study and existing peer-reviewed literature are difficult given the limited published research regarding coal mine fire smoke exposures. Previous research conducted by the HHS showed an association between the dispensing of respiratory medications in the Latrobe Valley and mine fire-related PM_2.5_ smoke exposure [16] and some comparison is possible with other studies that have investigated wildfire PM smoke exposures. Associations were found between wildfire PM_10_ exposure following an American wildfire and survey-reported respiratory symptoms in children [31]. An Australian study found an increased risk of emergency department attendance for asthma during a 2006/2007 wildfire [10]. Several Canadian studies have found associations between wildfire PM_2.5_ and asthma-related physician visits [32,33,34], hospital admissions [32] and the dispensing of the reliever medication salbutamol [33,34,35]. 

Cough is a physiological mechanism to clear inhaled particles from the respiratory tract and phlegm an indicator of mucus production, so the associations found between PM_2.5_ exposure and these symptoms are plausible. The finding that wheeze was more strongly associated with peak exposure compared to mean exposure may be because residents more prone to respiratory conditions were more likely to take protective action, such as relocating away from the fire [31] or increasing inhaled medications. Mine fire-related PM_2.5_ concentrations peaked on day two of the fire [8], at which point residents may still have been residing at their Morwell residential or work address and their peak exposure value would reflect this. If participants subsequently relocated, they would have a reduced mean exposure score, making it more difficult to identify an association.

Stronger dose response relationships were observed in males and those aged 18–65 years. Possibly, these groups were more active outdoors and/or employed during the mine fire and had less opportunity to relocate outside the exposure zone. Additionally, there may be an element of survivor bias [36] in the older age group. Relocation may also explain why gender stratification found that chronic cough in females was more strongly associated with peak compared to mean exposure, as women may have had more flexibility to relocate. The stronger OR between PM_2.5_ and asthma since 2014 in males may reflect a combination of the mine fire triggering the diagnoses of previously unrecognised asthma and/or small number instability (n = 18) as reflected in the wider CIs (OR 1.58 (1.10 to 2.29; 0.014)).

There is mixed evidence in the literature regarding the age group most susceptible to PM-related respiratory symptoms. Stronger associations were found in the 20–34 years group, relative to other ages, for the dispensing of respiratory medications following the Hazelwood mine fire [16]. Henderson [32] found stronger associations for middle-aged adults relative to older adults for respiratory physician visits following a wildfire. However, other wildfire studies found the relative strength of associations for different age groups varied for different respiratory outcomes [11,37]. Gender stratification has generally found that women have stronger associations than males. Following the mine fire, we found that women had a slightly stronger association than men for the dispensing of respiratory medication [16]. Following wildfires, an increased risk of hospital admissions for asthma in females was reported by Delfino [11] and mixed results for different respiratory-related emergency department visits were found by Tinling [38].

Building materials of participants’ residences were associated with respiratory outcomes. Residences constructed from brick/concrete were protective, possibly because relative to weatherboard houses, there was reduced penetration of PM_2.5_. Residences with roofs constructed from tin/metal were associated with increased ORs, possibly because in the study area, houses with tin roofs were often constructed from weatherboard and houses with tile roofs from brick. While metal roofs may have reduced PM_2.5_ penetration rates relative to tiles, the weatherboard house construction would have increased penetration. 

Wildfire studies have found that particulate penetration rates are higher for homes with reduced air-tightness [39,40] and that reducing indoor PM_2.5_ concentrations with air-filters reduces respiratory symptoms [41]. There may also have been some inter-relationship with socio-economic factors. Weatherboard houses were generally less expensive and not always well maintained and studies have found that lower socio-economic groups may be more susceptible to wildfire smoke exposure [12,42]. The two-way interaction found between PM_2.5_ and housing materials for shortness of breath may be a combination of socio-economic factors, pre-existing health conditions and possible indoor exposure differences. Nocturnal shortness of breath can be an indicator of asthma or heart failure and resting shortness of breath an indicator of severe COPD, and those from lower socio-economic groups may be more prone to illness. While our study controlled for known socio-economic and health confounders, it is possible that some residual confounding remained. 

The prevalence of self-reported asthma in our study was relatively high compared to the national Australian average (26% versus 11.2%) [43], possibly indicating high levels of existing poor respiratory health in participants, and/or a degree of selection bias. Other studies of communities living in close proximity to surface coal mining have also found an increased risk of poor health outcomes, including respiratory disease, which could not be explained by traditional risk factors such as age, occupational exposure or socio-economic status [44]. Over time, residents were exposed to high background levels of fine and ultrafine particulates, generated as a result of surface coal mining operations, such as diesel fumes generated during coal transportation [17].

These study results contribute to health policy responses in the event of future mine fire or wildfire pollution episodes. Study participants included both males and females and adults of all ages, and therefore, the results would have external validity in other similar communities affected by pollution episodes. In addition to respiratory outcomes, the HHS has shown that increments in PM_2.5_ exposure are associated with poorer mental health, in particular, with increments in trauma-related distress such as intrusive thoughts [19]. Further research is underway investigating possible comorbidity between respiratory and mental health in this cohort. Future research will also investigate any association between mine fire smoke PM_2.5_ exposure and cancers.

### Strengths and Limitations

A major strength of our study was the ability to gather individual-level health data, confounding factors, including residence construction materials, detailed hourly mine fire-related PM_2.5_ concentrations and time-location dairies for each participant. Additionally, the use of individual exposure scores rather than community level exposure measurements may have reduced the risk of exposure misclassification and increased the power of the study to identify associations between health and exposure [45]. However, some uncertainties were associated with the use of modelled PM_2.5_ concentrations, such as assumptions regarding emission characteristics, coal burn depth, local wind conditions and smoke plume dispersion [8]. Furthermore, details regarding housing material and roofing type were only collected for the principal residence of each participant and the proportion of time spent at this location would have varied between participants. 

There was a 26-month lag between the end of the mine fire (March 2014) and the commencement of recruitment (May 2016), which may have led to some recall bias. Most respiratory symptom questions referred to the previous 12 months, which would, at minimum, have been 14 months after the fire period. Therefore, influences other than the mine fire may have contributed to symptom reporting. Alternatively, it is possible that the reporting of respiratory symptoms after an extended period post the mine fire may be an indicator of persistent or long-term impacts of the event. Respiratory outcomes may also have been overestimated due to comorbidity between respiratory symptoms and stress. Traumatic stress has been associated with the activation of a number of neurobiological systems, including inflammatory cytokines [46], which may, in turn, impact on airways reactivity [47,48] and therefore, stress may have presented as respiratory outcomes.

Limited self-reported data were collected regarding participants use of medications to control respiratory symptoms. A binary “yes” or “no” response was collected regarding the use of inhaled steroids in the “last 12 months”. Details about steroid inhaler medication quantity, frequency and duration of use were not collected. Due to the cross-sectional nature of the study, the sequence of medication use and symptoms was unknown. Hence, inhaled steroid use was not controlled as a confounding variable, as it resembled more of a marker for underlying asthma or COPD, which were already controlled in the model. The sensitivity analysis found little or no difference when steroid use was fitted as a variable to the multivariate models for both mean and peak mine fire-related PM_2.5_. However, further research is underway which includes more detailed data about respiratory medication use in a sub-sample of this cohort.

## 5. Conclusions

This study found associations between increments in coal mine fire-related PM_2.5_ smoke exposure and increases in respiratory symptoms in adults, about 2.5 years after the event. The strongest association observed was between mean mine fire-related PM_2.5_ exposure and chronic cough. Mean PM_2.5_ exposure was also associated with chronic phlegm and peak PM_2.5_ exposure was associated with chronic cough and current wheeze. Males, participants aged 18-64 years and participants living in homes constructed from non-brick/concrete materials (largely weatherboard) or homes with tin/metal roofs had higher risk. These findings contribute to the formation of public health policy responses in the event of future major pollution episodes.

## Figures and Tables

**Figure 1 ijerph-16-04262-f001:**
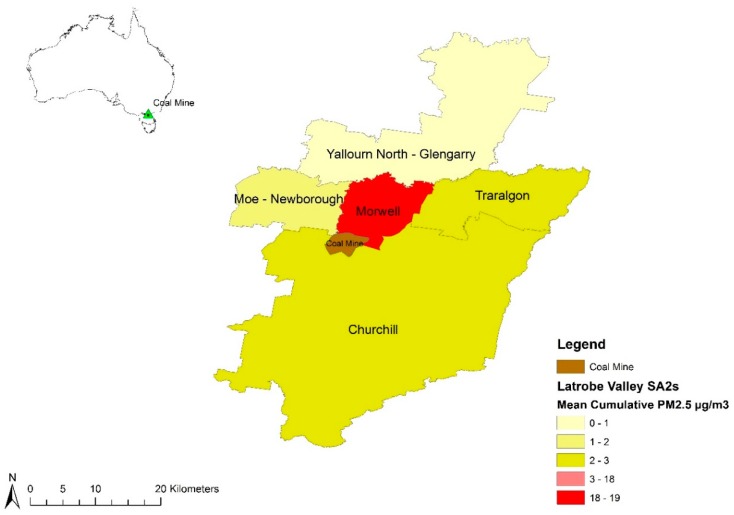
Map of the coal mine and the five statistical areas (SA2s) comprising the Latrobe Valley exposure zone. Shading reflects cumulative mean 24-h mine fire-related PM_2.5_ concentrations by SA2, for the period 9 February–31 March 2014. Note: Insert top left indicates the location of the mine within Australia.

**Figure 2 ijerph-16-04262-f002:**
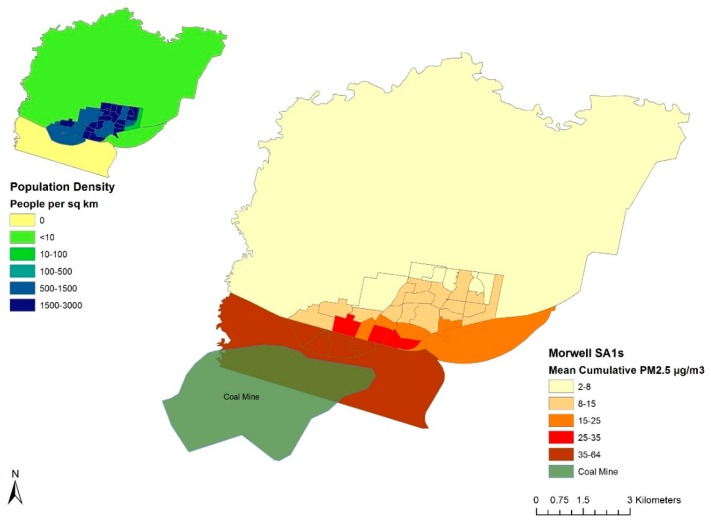
Map of the statistical areas (SA1s) within the Morwell SA2. Shading reflects cumulative mean 24-h mine fire-related PM_2.5_ concentrations by SA1, for the period 9 February–31 March 2014. Note: Insert top left displays 2014 population density for each Morwell SA1.

**Figure 3 ijerph-16-04262-f003:**
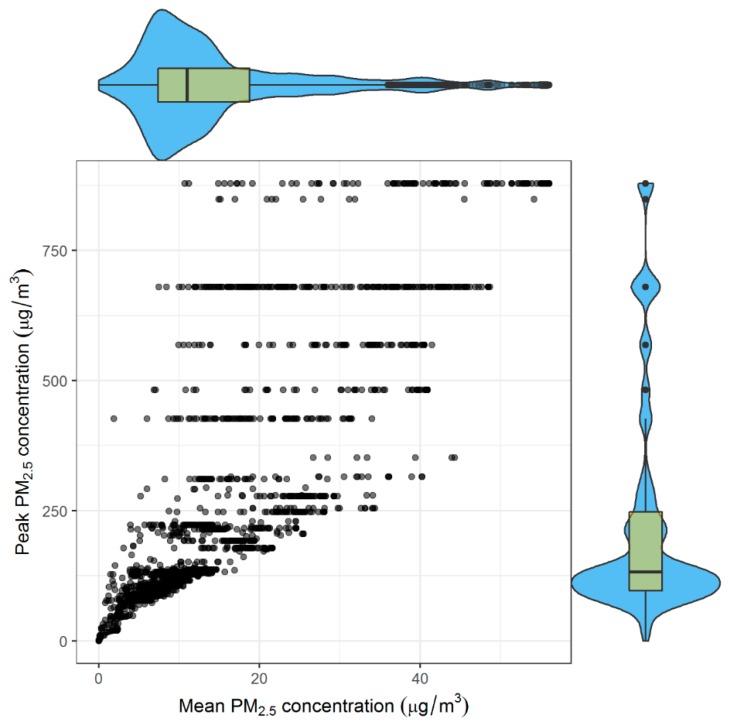
Scatterplot and Violin probability density plots of participants’ mean cumulative 24-h and peak 12-h mine fire-related PM_2.5_ concentrations; Note: Scatterplot (located centre) plots peak versus mean mine fire-related PM_2.5_ concentration values for each participant. Violin probability density plots for mean 24-h PM_2.5_ and peak 12-h PM_2.5_ are located top and right of the scatter plot respectively. Boxplot inserts indicate the 25th and 75th percentiles. The horizontal line indicates median. Whiskers indicate lowest and highest values, excluding outliers. The dots indicate outliers. The violin shapes surrounding the box plots are rotated kernel density estimations, indicating the full distribution of PM_2.5_ concentrations. There are two identical distributions per box plot; one mirrored on either side. The thicker sections of the violin shape represent higher frequencies.

**Figure 4 ijerph-16-04262-f004:**
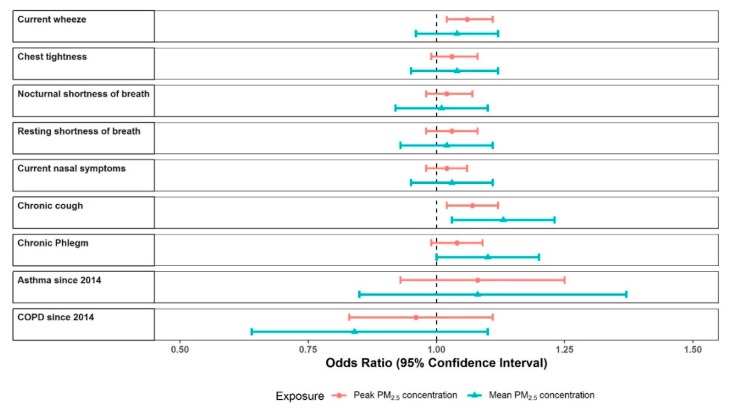
Forest plot of multivariate model results for associations between respiratory outcomes and mine fire-related PM_2.5_ concentrations.

**Figure 5 ijerph-16-04262-f005:**
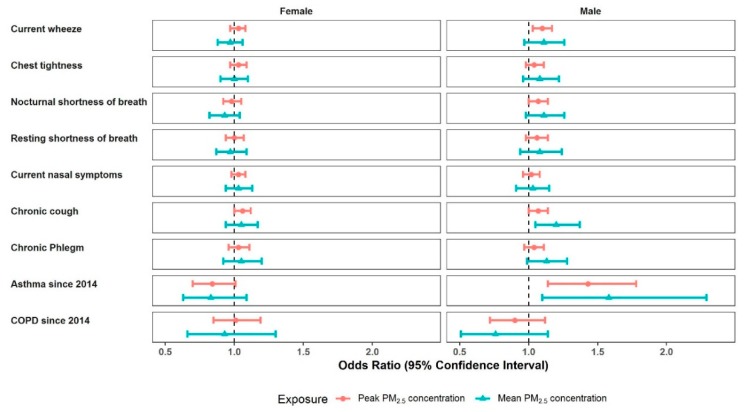
Gender stratified forest plots (female and male groups) of multivariate model results for associations between respiratory outcomes and mine fire-related PM_2.5_ concentrations.

**Figure 6 ijerph-16-04262-f006:**
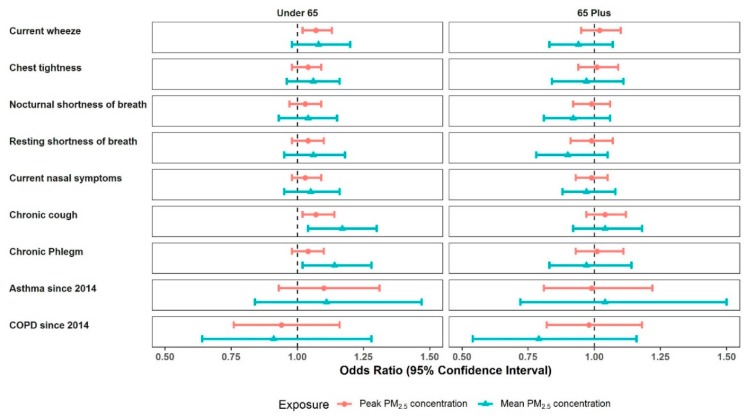
Age stratified forest plots (under 65 years and 65 years plus groups) of multivariate model results for associations between respiratory outcomes and mine fire-related PM_2.5_ concentrations.

**Table 1 ijerph-16-04262-t001:** Summary of predictor variables across tertile categories of Morwell participants’ mean 24-h mine fire-related PM_2.5_ concentrations (*n* = 3096).

Predictor Variables	Total	Low Exposure	Medium Exposure	High Exposure	*p*-Value *
	Weighted Mean (SD)	Weighted Mean (SD)	Weighted Mean (SD)	Weighted Mean (SD)	
Age during the mine-fire	48.07 (18.59)	47.71 (18.68)	46.69 (17.88)	49.78 (19.05)	0.020
	***n* (weighted %)**	***n* (weighted %)**	***n* (weighted %)**	***n* (weighted %)**	***p*-value ***
Male	1389 (48%)	447 (47%)	461 (50%)	481 (47%)	0.371
Employment					0.337
Paid employment (FT, PT, self-employed)	1311 (51%)	428 (52%)	441 (53%)	442 (50%)	
Other (student/volunteer/home-duties/retired)	1368 (35%)	464 (36%)	430 (33%)	474 (37%)	
Unemployed	139 (6%)	50 (6%)	48 (7%)	41 (5%)	
Not working due to ill-health	239 (7%)	67 (6%)	82 (7%)	90 (8%)	
Highest educational qualification					0.082
Secondary up to year 10	1006 (27%)	317 (26%)	357 (29%)	332 (26%)	
Secondary year 11–12	668 (24%)	203 (22%)	236 (27%)	229 (23%)	
Certificate (trade/apprenticeship/technicians)	996 (34%)	355 (35%)	291 (32%)	350 (35%)	
University or other Tertiary Institute degree	385 (15%)	136 (17%)	113 (12%)	136 (15%)	
Married/Defacto	1852 (57%)	676 (61%)	601 (56%)	575 (54%)	0.019
Smoking status					0.027
Never	1495 (51%)	516 (54%)	493 (52%)	486 (47%)	
Former smoker	1052 (31%)	356 (30%)	328 (29%)	368 (33%)	
Current smoker	516 (18%)	141 (15%)	183 (19%)	192 (20%)	
Work exposure					0.098
Not exposed	1825 (62%)	620 (64%)	611 (64%)	594 (59%)	
Coal mine/station exposed	494 (14%)	162 (13%)	147 (12%)	185 (16%)	
Exposed, but not coal mine/station	777 (24%)	244 (22%)	254 (24%)	279 (25%)	
Roof main material—iron/tin	977 (32%)	231 (24%)	346 (34%)	400 (37%)	<0.001
Home main material—concrete/brick	1927 (61%)	774 (76%)	606 (58%)	547 (51%)	<0.001
Asthma pre 2014	715 (26%)	248 (27%)	238 (26%)	229 (24%)	0.259
COPD pre 2014	148 (4%)	44 (3%)	45 (3%)	59 (4%)	0.379

* Weighted chi-square test for proportion differences across exposure categories.

**Table 2 ijerph-16-04262-t002:** Summary of outcome variables across tertile categories of participants’ mean mine fire-related PM_2.5_ concentrations.

Outcome Variables	Total	Low Exposure	Medium Exposure	High Exposure	*p*-Value *
	*n* (Weighted %)	*n* (Weighted %)	*n* (Weighted %)	*n* (Weighted %)	
Current wheeze	1317 (42%)	422 (41%)	418 (40%)	477 (46%)	0.021
Chest tightness	792 (27%)	237 (25%)	265 (27%)	290 (29%)	0.201
Nocturnal shortness of breath	635 (20%)	203 (20%)	207 (19%)	225 (21%)	0.604
Resting shortness of breath	611 (20%)	194 (19%)	197 (20%)	220 (22%)	0.397
Current nasal symptoms	1358 (44%)	445 (44%)	425 (41%)	488 (47%)	0.114
Chronic cough	989 (31%)	296 (26%)	340 (32%)	353 (34%)	0.004
Chronic phlegm	785 (25%)	242 (23%)	259 (25%)	284 (27%)	0.133
Asthma since 2014	59 (2%)	21 (2%)	18 (2%)	20 (2%)	0.722
COPD since 2014	60 (1%)	21 (1%)	20 (2%)	19 (1%)	0.984

* Weighted chi-square test for proportion differences across exposure categories.

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
