# Peer review of "Associations between Respiratory Health Outcomes and Coal Mine Fire PM2.5 Smoke Exposure: A Cross-Sectional Study"

_ijerph, 2019, doi:10.3390/ijerph16214262_

Round 1
Reviewer 1 Report
The article entitled “Associations between respiratory health outcomes and coal mine fire PM2.5 smoke exposure: a cross sectional study” is interesting and publishable in the journal after considering some changes.
-The introduction is too short to elaborate state of the art. Authors only focused on in Australia in the introduction; however, health outcome due to pollution is serious problem around the globe. It is required to write about previous studies around different continents in the introduction and write a precise state of the art in the introduction section.
I suggest to author to review and cite recent studies for examples
“Human health damages related to air pollution in China. Environmental Science and Pollution Research, 1-11”
- I suggest to extend conclusion part and write down a short summary of results .
-Figures have illustrated clearly. The data presented in tables is clear and understandable.
Reviewer 2 Report
The pre-existing rates of asthma in the population seem a little high. The authors should comment on that and any potential reason they may know about why that may be the case. Reference 12 is appropriate for that purpose but not sufficient. One is left to wonder whether the post-fire time period change in symptoms could be attributable to that underlying exposure for the increased asthma rates. One of Hendryx's earlier papers would seem to indicate truck traffic and its accompanying ultrafine particulate exposure may be responsible. One could certainly attribute the symptom changes described in this paper to a similar (and longer duration) cause around this coal mining area. If it cannot be dispelled completely as a cause, it should at least be a little more directly acknowledged.
Reviewer 3 Report
Thank you for this interesting and well-written manuscript on association between respiratory health outcomes 2 and coal mine fire PM2.5 smoke exposure.
The paper is well written and easy to read.
Although the present data are intriguing, there are several issues that should be addressed:
Please add in abstract short conclusion Can you present another comorbidities not only asthma and COPD or if you have data on Charlson index? Do you have data on medication of these persons, especially with COPD or asthma?Author Response
Please see the attachment
